



# Merged Observatory Data Files (MODFs): An Integrated Observational Data Product Supporting Process-Oriented Investigations and Diagnostics

Taneil Uttal[1,a,*], Leslie M. Hartten[2,1,a,*], Siri Jodha Khalsa[2,3], Barbara Casati[4], Gunilla Svensson[5], Jonathan Day[6], Jareth Holt[5], Elena Akish[1,2,b], Sara Morris[1,7], Ewan O'Connor[8], Roberta Pirazzini[8], Laura X. Huang[9], Robert Crawford[9], Zen Mariani[9], Øystein Godøy[10], Johanna A.K. Tjernström[11,10], Giri Prakash[12], Nicki Hickmon[13], Marion Maturilli[14] and Christopher J. Cox[1]

[1]Physical Science Laboratory, National Oceanic and Atmospheric Administration, Boulder, CO 80305-3328, USA
[2]Cooperative Institute for Research in the Environmental Sciences, Boulder, CO 80309, USA
[3]National Snow Ice and Data Center, Boulder, CO 80309-0449, USA
[4]Meteorological Research Division, Environment and Climate Change Canada, Dorval, QC H9P-1J3, Canada
[5]Department of Meteorology and Bolin Centre for Climate Change, Stockholm University, 106 91 Stockholm, Sweden
[6]European Center for Medium-Range Weather Forecasts, Reading, RG2 9AX, United Kingdom
[7]Global Monitoring Laboratory, National Oceanic and Atmospheric Administration, Boulder, CO 80305-3337, USA
[8]Finnish Meteorological Institute, Helsinki, FI-00101, Finland
[9]Meteorological Research Division, Environment and Climate Change Canada, Toronto, ON M3H-5T4, Canada
[10]Norwegian Meteorological Institute, Olso, N-0313, Norway
[11]Swedish Meteorological and Hydrological Institute, Norrköping, SE-60176, Sweden
[12]Department of Energy, Oak Ridge National Laboratory, Oak Ridge, TN 37830, USA
[13]Department of Energy, Argonne National Laboratory, Lemont, IL 60439, USA
[14]Alfred Wegener Institute Helmholtz Centre for Polar and Marine Research, Potsdam, D-14473, Germany
[a] retired
[b] currently at: Broomfield, CO 80020, USA

*Correspondence to*: Taneil Uttal (taneil.uttal@gmail.com), Leslie M. Hartten (Leslie.Hartten@colorado.edu)



**Abstract.** A large and ever-growing body of geophysical information is measured on campaigns and at specialized observatories as a part of scientific expeditions/experiments. These collections of observed data include many essential climate variables (as defined by the World Meteorological Organization), but are often distinguished by a wide range of additional non-routine measurements that are designed to not only document the state of the environment, but also the drivers that contribute to that state. These field data are not only used to further understand the environmental processes through observation-based studies, but also to provide baseline data to test model performance and to codify understanding to improve predictive capabilities. To address the considerable barriers and difficulty in utilizing these diverse and complex data for observation-model research, the Merged Observatory Data File (MODF) concept has been developed. The MODF combines measurements from multiple instruments into a single file that complies with well-established data format and metadata practices and has been designed to parallel development of corresponding Merged Model Data Files (MMDFs). Using MODF and MMDF protocols will facilitate the evolution of Model Intercomparison Projects into Model Intercomparison and Improvement Projects by putting observation and model data 'on the same page' in a timely manner. The MODF concept was developed especially for weather forecast model studies in the Arctic. The surprisingly complex process of implementing MODFs in that context refined the concept itself. Thus this article explains the concept of MODFs by providing details on the issues that were revealed and resolved during this first specific implementation. Detailed instructions are provided on how to make MODFs, and this article can thus be considered a MODF creation manual.

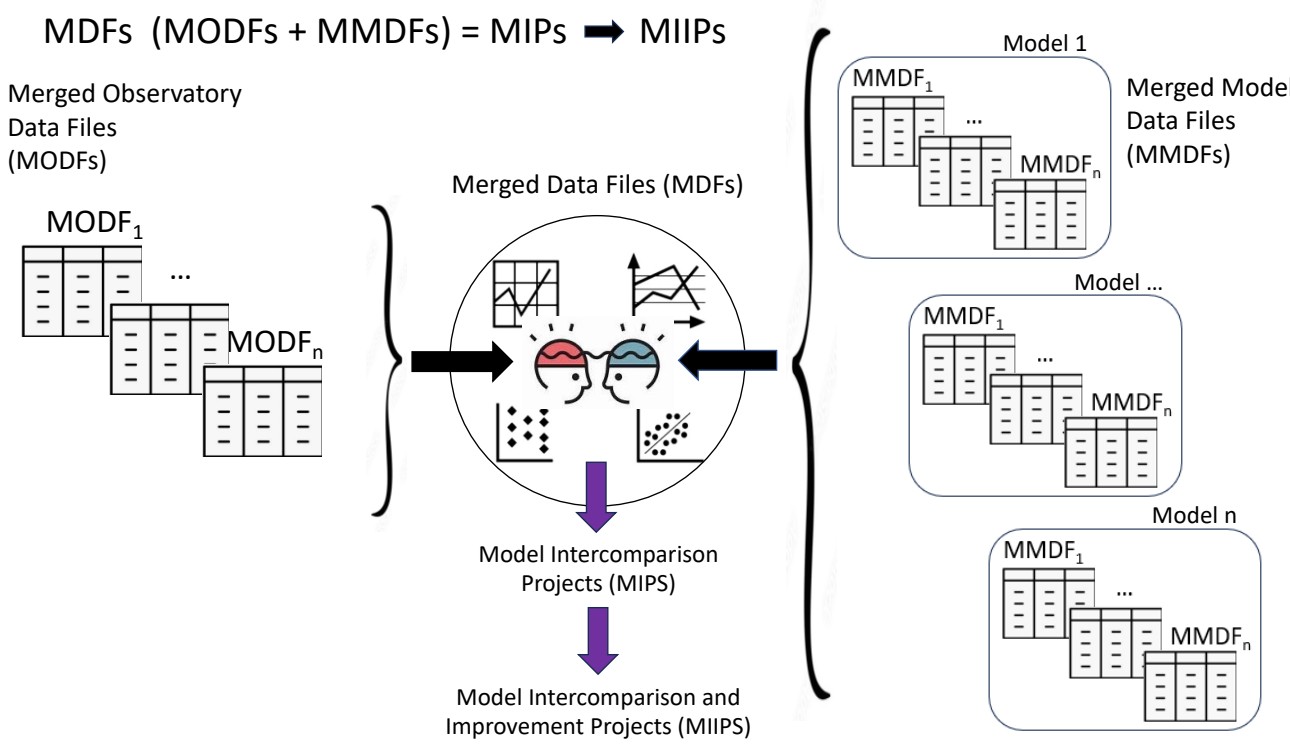





# 1 Introduction

The Merged Observatory Data File (MODF) concept is based on the simple principle of combining measurements made by multiple, co-located instruments from research observatories and campaigns into a single file that complies with already established data stewardship standards. 'Observatory' here refers to a facility that measures an extensive inventory of collocated geophysical variables that have been chosen with the intention of investigating specific, usually interrelated, physical processes in order to answer hypothesis-driven science questions. In this context, an observatory could be a land site

or a research vessel. While it is standard scientific operating procedure to co-locate research grade instruments both continuously at observatories and episodically for field campaigns (often side-by-side with routine operational station instruments with long operational histories) there are generally no standard procedures for coordinated data management such as those that have been developed for operational data. Thus, the data from separate instruments can be scattered between separate files with different authors, formats, metadata, physical archive locations, and use restrictions.


The specific MODF realization presented here is for Arctic observatories and field campaigns that resulted from initiatives established during the Year of Polar Prediction (YOPP; Goessling et al., 2016; Jung et al., 2016; Wilson et al., 2023). One key YOPP activity was the YOPP supersite Model Intercomparison Project (YOPPsiteMIP; Day et al., submitted), which was designed to facilitate process-based validation of numerical weather prediction (NWP) models at polar locations during

Special Observing Periods (SOPs). The concept of MODFs and their forecasting analogs, Merged Model Data Files (MMDFs), was motivated by the YOPPsiteMIP community's desire to have the same variables from observations and models in easy-to-use files of the same structure in order to explore small-scale parameterized processes that are not represented well in the forecast models. These MODFs thus provide an integrated observation database to support model process representation through parameterization improvements for weather forecasts in the polar regions. At the same time,

they also facilitate comparative observational studies across Arctic sites.

The MODF concept addresses the problem that research-grade, process-level observations are currently underutilized for model evaluation of parameterization deficiencies. As weather forecasting models increase in complexity and include detailed representation of land, ocean, ice, and snow in addition to the atmosphere, it is increasingly important to evaluate

processes using observations recorded through the whole earth system column, including the fluxes at their interfaces, to inform model development. This requires multi-variate process-oriented diagnostic methods utilizing data that spans these multiple components to understand model error. An essential component of the MODF concept is that analagous Merged Model Data Files (MMDFs) can be developed with extracted model data near and around the observatory sites. Together, these are defined as Merged Data Files (MDFs). The MDFs, which bring together observations from different earth system

components as well as model output in a standard file format, provide the basis for this and will support Model Intercomparison and Improvement Projects (MIIPs) as an evolution from Model Intercomparison Projects (MIPs). The idea





of MIIPs (a new acronym that we define here) does not imply that previous MIPs did not have productive results. However, through the development of matched MODF and MMDF data sets, it is implied that what have often been intercomparison studies can be smoothly extended to model improvements.

## 2 Background

Using observations and model outputs symbiotically is an area of ongoing effort and research with recognized challenges (Holtslag, et al., 2013). Many levels of MIPs have been in progress for decades (Stephens et al., 2023). In addition, existing efforts and methodologies facilitate the usage of increasingly heterogenous data sets in model–observation fusion efforts through assimilation (Gettleman et al., 2022) and for inputs into multivariate artificial intelligence analyses (Boukabara et al., 2021). There are well-organized systems for managing the data from operational surface networks, upper air networks and satellites that are uploaded into the Global Telecommunications System (GTS) that is overseen by the World Meteorological Organization (WMO) (see Global Telecommunications System (GTS), 2023; WMO, 2020). However, GTS data are only readily available directly to national forecasting centers (which presumably have developed institutionally specific reading and ingesting routines) and via products developed by WMO institutional repositories[1] (Bojinski et al., 2014; Lavergne et al., 2022). The latter have necessarily gone through various quality control, formatting, averaging and sometimes interpolation to create globally uniform products. As a result of the standardized processing, it is likely that information on high-resolution, rapid and extreme events (Sardeshmukh et al., 2015) may have sometimes been lost.

The MODF schema has been specifically developed for managing observatory and campaign research observations as opposed to operational observations. Research observations target local-scale and often rapid or extreme processes that are intended to lead to discovery of the physics within the atmosphere as well as the physics that govern the coupling processes between the atmosphere and the underlying surface. The surface can be land, ocean, ice or any of the three, often with obfuscating layers of plant and/or snow cover that are themselves components of the system and separate objects of study. Research-grade data pose many additional challenges regarding data latency, accessibility and uptake issues, and institutional ownership, compared to data that have been managed more systematically specifically for operational purposes.

The data science community is aware of these issues and in response has developed FAIR (Findable, Accessible, Interoperable, Reusable) data principles (Wilkinson et al., 2016) that can be applied to individual data sets. However, Wilkinson et al. explicitly stated that "These high-level FAIR Guiding Principles precede implementation choices, and do not suggest any specific technology, standard, or implementation–solution; moreover, the principles are not, themselves, a standard or a specification." Whereas many FAIR solutions are implemented by web services (e.g., Buck et al., 2019), the MODF concept described here can be considered an alternative: an integrated data product that is based on the same

---

[1] https://climatedata-catalogue.wmo.int/homepage



metadata conventions that have been developed for web service solutions. The considerations and steps described here for creating MODFs can therefore be considered a particular FAIR implementation–solution specifically for observatory and
110   campaign data.

## 3 The MODF Concept

Figure 1 is a conceptual schematic of the end-to-end process involved with data collection, data quality control (QC) and processing, metadata information collection and data amalgamation into netCDF files (Unidata, 2023) that follow the NetCDF Climate and Forecast (CF) Metadata Conventions (Eaton et al., 2022, hereafter the CF Metadata Conventions). The
115   process for turning model forecast output into MMDFs is qualitatively similar[2], including the use of a particular set of global and variable attributes, since the specifications were developed by modellers as well as observationalists.

---

[2] Differences generally arise from the fact that forecast models can produce tendencies on timestep scales that are typically not available from field instruments, and that model output is typically regularly distributed in time and space rather than coming from discrete instruments which may move irregularly in space, which operate at different optimized cadences, and which are subject to physical or power disruptions.





**Figure 1: Instruments, measurements, data processing and metadata generation for (A1) Surface Atmospheric State, (A2) Broadband Surface Radiation, (A3) Surface Turbulent Fluxes, (A4) Upper Atmospheric Profiles, (A5) Terrestrial Sub-Surface and (A6) Ocean and Sea Ice/Snow collected into netCDF files with specified (B) Global Attributes and (C) Variable Attributes that are compliant with FAIR principles.**

The different types of geophysical data (A1 thru A6) that typically can compose observatory and campaign research data are described below.



### 3.1 Surface Atmospheric Measurements

Surface atmospheric measurements at observatories or during campaigns are made with research-grade thermometers, hygrometers and anemometers, which often measure at higher frequency and with more carefully calibrated sensors than operational weather stations. Such observations are sometimes side-by-side with operational weather service stations, providing context to multi-decadal operational records. Research meteorological measurements are frequently redundant, deployed at multiple locations across a site (on a scale smaller than an NWP model grid cell) or at different levels on towers to get detailed profiles in the near-surface boundary layer. Resulting variables measured are temperature, pressure, relative humidity, and the eastward and northward components of the windspeed. Both operational and research measurements of these atmospheric state variables can be considered to have well-quantified uncertainties determined by instrument calibration and tolerances.

Broadband surface radiation is measured by radiometers that measure incoming and outgoing, shortwave and longwave radiation and associated variables such as surface temperature, sky brightness temperature and derived variables such as albedo. There are multiple commercial and experimental instrument options for measuring broadband radiation, and in the Arctic a particularly wide range of methods must be applied to keep glass domes clear of obstructions (e.g., ice, snow, dust or sea-salt) such as heating, ventilation and manual cleaning (Cox et al., 2021). The Baseline Surface Radiation Network (Ohmura et al., 1998) is a global repository for standardized radiation products that are traceable to the WMO World Radiation Radiometric Reference. However, BSRN is not designed to accommodate short-term campaign data sets and does not account for different methods for data QC and processing (Matsui et al., 2012; Long and Shi, 2006; Long and Shi, 2008).

Surface Turbulent Flux variables are calculated and inferred by a number of different methods. Eddy-correlation techniques (Kaimal and Finnigan, 1994) are based on measurements by fast-response sonic anemometers and hygrometers with built-in fast-response temperature sensors. It is often necessary to make considerable site-specific adjustments to processing methods accounting for local surface roughness, sensor height, and obstructed wind-direction sectors. In the polar regions, sensors operate near the thresholds of instrument ratings for detection and environmental conditions and have to be quality-checked for periods of riming. There are specific challenges for the cold snow- and ice-covered surface conditions over land (Grachev et al., 2018) and over the icepack of the central Arctic Ocean (Andreas et al., 2010; Cox et al., 2023). Eddy-correlation methods are only valid under stringent environmental conditions, which results in frequent data gaps. Alternatively, turbulence variables can be calculated by bulk-aerodynamic methods (Monin and Obukhov, 1954; Mahrt and Sun, 1995), but results are not as meaningful for comparing to models as it becomes a model-to-model rather than a model-to-observation comparison. Commercial packages have been developed and compared for calculating latent, sensible and gas fluxes (Mauder et al., 2008; Fratini and Mauder, 2014), but caution must be exercised in using results and it is important to understand the basis on which variables are derived by proprietary commercial or custom software. The resulting flux





variables are latent and sensible heat fluxes, friction velocity, surface stress (momentum flux), drag coefficient, kinematic temperature scale, Monin-Obhukov stability parameter and dissipation rate of the turbulent kinetic energy. Given the variety of physical conditions and complex methodologies for calculating turbulent fluxes, an assessment of the interoperability and consistency between flux products from different data collections is still necessary despite the existence of operational flux data networks such as FLUXNET (Baldocchi et al. 2001) and AmeriFlux (Baldocchi et al. 1996; Boden et al. 2013).

Solid and liquid precipitation are both measured at the surface but are notoriously difficult to characterize accurately. Precipitation accumulation with hourly to 6-hour frequency is traditionally measured by gauges within a surface precipitation network, usually operated by operational weather centers or hydrological agencies. Measuring solid precipitation presents a number of unique challenges; snow pillows and the Double-Fence Automated Reference (DFAR) configuration around gauges provide reliable estimations of precipitating snow. However, snow precipitation measurements from WMO standard installations with Single-Alter-shielded and unshielded gauges are affected by the undercatch of solid precipitation in windy conditions (Nitu et al., 2018; Kochendorfer et al., 2022). The WMO Solid Precipitation Intercomparison Experiment (SPICE) analyzed this undercatch and developed adjustment functions for correcting it (Kochendorfer et al., 2018; Wolff et al., 2015) which are now being used in verification practices (Køltzow et al., 2020; Buisan et al. 2020, Casati et al., 2023).

## 3.2 Upper Atmospheric Profiles

Radiosonde data are typically treated as if they are instantaneous vertical profiles of the troposphere by collapsing the balloon-borne trajectory of the instrument package into a profile that is time indexed with the launch time. The measurements of temperature, dewpoint temperature, pressure, humidity, and winds are high cadence (on the order of seconds) during the balloon ascent with individual launches being low cadence (every 12 hours standard launch intervals at 00 and 12 UTC). During intensive campaign periods, the launch frequency is often increased to 4–6 sondes per day. A full sounding can take 2 hours to ascend and can travel up to 200 km horizontally, and important fine-scale information is available if full, original trajectory information (time–height/pressure level–latitude–longitude) coordinates are maintained to assess the spatial displacement.

Many observatories and campaigns support the operation of radars, lidars, sodars, profilers and microwave radiometers which separately and in combination can remotely infer properties through the depth of the planetary boundary layer (PBL) and free atmosphere. The systems use active (transmission and interpretation of reflected signal) and passive (detection of natural atmospheric signal) sensing techniques. Through a significant body of research on retrieval methods, the systems can determine properties such as cloud base, cloud liquid water path, cloud ice water path, cloud liquid water content, cloud ice water content, hydrometeor sizes and shapes, snowfall rates (Matrosov et al. 2022), degree of riming, aerosol extinction





coefficients, winds, temperature and humidity. These advanced products may be obtained from diverse instrument hardware configurations and technologies as well as site-specific scanning and collection schedules. Robust site-independent retrieval methodologies that are consistent across networks such as the products produced by Cloudnet (Illingworth et al., 2007) and the ARM Active Remote Sensing of Clouds product (Clothiaux et al., 2001) have been developed, however, they may not be
consistently implemented.

## 3.3 Terrestrial Surface and Subsurface

The terrestrial surface and subsurface is composed of complex layers of different soil, rock, ice (permafrost) and vegetation layers that can be covered with surface snow and ice with evolving density, heat capacity, conductivity and chemistry. Thermistor strings can be co-located with tenslometers; the variables measured are gradients of temperature and moisture
respectively with thermal conductivity and heat capacity determined as fixed intrinsic properties from soil samples. Thermistor strings can also be co-located with moisture probes, providing vertical profiles of soil temperature and moisture respectively down to depths ranging from 0.1 to 2 m depending on the soil type. Snow depth, like precipitation, is a difficult quantity to measure representatively; techniques vary from using simple measurement stakes to downward-looking mast- or tower-mounted accoustic devices.

## 3.4 Ocean, Sea Ice, and Snow Surface and Subsurface

Ocean thermistor measurements are accompanied with conductivity and pressure measurements with resulting variables determined for salinity, temperature and depth. Additional current meters allow determination of turbulent fluxes using eddy correlation techniques similar in principle to those used for atmospheric fluxes. Measurement of sea ice and snow macro- and microphysical properties below the atmosphere/ice or atmosphere/snow interface (i.e. subsurface) is increasingly
sophisticated with upward and downward looking acoustic devices on ice buoys determining ice thickness and snow depth (Zuo et al., 2018). Measurements are made of snow and ice density, crystalline structure and salinity via manual sampling.

## 4 MODFs for the Year of Polar Prediction

The World Meteorological Organization (WMO) Polar Prediction Project (PPP) organized the Year of Polar Prediction (YOPP; PPP Steering Group & Co-authors, 2019; Jung et al., 2016) which concluded in 2022 (Wilson et al., 2023). The
YOPPsiteMIP (Svensson et al., 2020; Day et al., submitted) working group envisioned matched sets of observation and model data to support model process diagnostics. The efforts of this working group resulted in a metadata schema for MODFs and MMDFs (collectively known as MDFs) as well as an iterative production workflow. The YOPPsiteMIP effort focused on Polar terrestrial stations, but because it was anticipated that a similar strategy would be applied to the YOPP endorsed Multidisciplinary drifting Observatory for the Study of Arctic Climate (MOSAiC) expedition (Shupe et al., 2022,



Nicolaus et al., 2022), the schema was developed to also accommodate expected additional ocean and sea ice observations
       from ships and on-ice platforms in the central Arctic Ocean.

       The first steps in MODF production are to assemble the available data files that will feed into the MODF, extract the desired
       data, and acquire the corresponding metadata while noting that each individual instrument and instrument group requires
both unique quality control and processing together with attribution tracking to produce and document the provenance of the
       geophysical variables. Given the heterogeneity and often research-grade nature of campaign data, the amount and quality of
       metadata for different variables is likely to be inconsistent. Although required metadata are ideally harvested from the
       internally documented data files, it is likely that many data sets will need additional metadata that will require interviewing
       original data collectors. (For an example of this other than our own, see Papoutsoglou et al., 2023).  Once individual data and
any available metadata from individual instruments and instruments suites are created and or assembled, the merging process
       can begin.

       As we developed the schema and workflow, we identified the following challenges and solutions.

Semantics: Data semantics address the issue of the same variable being given different names drawn from multiple or ad hoc
       naming conventions. A significant part of the MODF solution has been the development of an extensive schema based on
       already existing vocabulary standards.

       Units: Variables' units are frequently absent entirely from data sets or expressed with nonstandard abbreviations (Hanisch et
al., 2022). The MODF solution is to associate each variable with recommended units, typically as identified in the CF
       Standard Name Table (2023).  These are meant to compatable with and generally recognized by the Unidata (2020)
       UDUNITS package (Eaton et al., 2022).

       Attribution: Perhaps the greatest MODF challenge is not technical but rather cultural.  When data for a single MODF
product come from instruments operated from multiple institutions and researchers, there are complex issues with
       acknowledging the original sources of the data (Pierce et al., 2019; Nature Editorial, 2022).  Data from campaigns or
       programs usually have multiple institutions and individual researchers involved, all of whom have different performance
       metrics for original research.  This often leads to official or implicit data embargos so that the researchers who collected the
       data will have the first opportunity to publish research results. The design of MODFs is responsive to these issues by
providing a high level of attribution metadata including links to original data and data producers, supporting copious data
       citations (Vannen et al., 2020).



Data Heterogeneity*: Measurement heterogeneity is generated by how data are collected and processed. The MODF solution is to document variable derivation as much as possible from original data metadata: type of instrument, calibration information, method of deployment, quality control and processing histories, and original licenses.

Inconsistent Cadences: Different variables are collected on a range of native cadences varying from Hz for the fast response sensors needed to record phenomena that can change on short time scales (typically in the atmosphere) to sensors for which it is sufficient to sample on hourly or even daily time scales (typically in the terrestrial sub-surface) with a wide range in-between (typically for the ocean and ice sub-surfaces). The MODF solution is for each variable to retain its native recording cadence and to minimize any temporal averaging beyond that necessary for sensible data processing. In addition, original data sets are not interpolated to fill data gaps.

Redundancy: It is common during campaigns or at intensive observatory sites for there to be multiple measurements of the same variable (e.g., temperature). There is important information contained in redundant measurements when evaluating how representative a variable is of site characteristics or when comparing the in-situ observations to model grid cells and satellite footprints to evaluate local variability. The MODF solution is to include as many redundant measurements as possible, maintaining high-accuracy location information including for example soil and vegetation characteristics that help define the observations' microclimates.

Processing Levels*: Research-grade data sets typically require unique quality control and often extensive post processing which rarely can be fully automated. Frequently the state variables (e.g. temperature) are readily available in near real-time, whereas more complex variables such as turbulent fluxes can have a wide range in availability depending on whether they are output by commercial software (Fratini and Mauder, 2014), calculated from bulk variables, or based on customized calculations for complex environments. The MODF solution is to include full processing, data quality and usability metadata.

Versions: Since it is common for observed data to have versions, the MODF concept accommodates multiple product releases. Examples of data versions are, raw (sometimes just voltages); minimally quality checked by automated elimination of extreme physically unrealistic outliers; subjectively curated for unusual and if possible correctable operational or environmental conditions; and different levels of processing to produce retrieved variables (those based on more direct measurements). Frequently, highly processed and certified data are available months or even years after the original collection period. In some cases, all processing levels of data are available and archived. Two obvious ways of handling the situation are to either replace original data with more processed data as it becomes available (Figure 2A) or to keep all versions of data (Figure 2B). MODFs' rich metatdata includes careful version tracking for individual variables which encourages the latter.



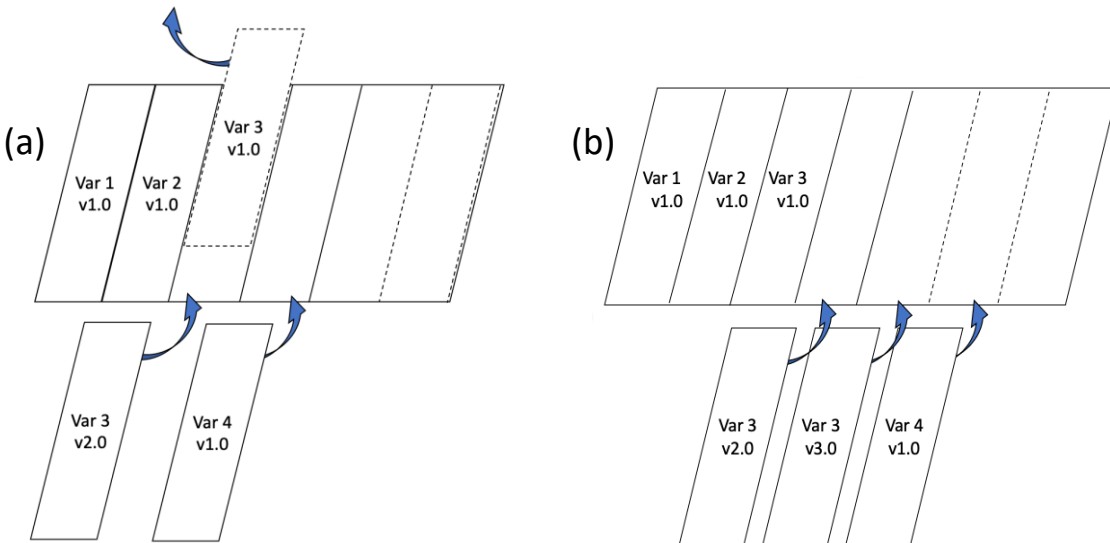

**Figure 2: There are 2 approaches for MODF augmentation (as new variables become available) and modification (as variables proceed through processing levels). In both approaches, variables that are immediately available (Var 1, Var 2) can be immediately ingested and variables can be added as they become available (Var 3, Var 4). The 2 approaches accommodate two strategies for variables that undergo post processing resulting in products with quality-control (QC) levels. In (a), original data (Var 3 v1.0) can be replaced with QC'ed data (Var3 v2.0). In (b), original data (Var3 v1.0) can be retained in the MODF and QC'ed data (Var3, 2.0) can be added.**

## 5 The H-K Variable Schema Table

The H-K Variable Schema Table, developed for the YOPPsiteMIP (Hartten and Khalsa, 2022, hereafter the H-K Schema), provides guidelines for creating both MODF and MMDFs as netCDF files (Unidata, 2022) with consistent variable names and metadata. Hereafter the discussion in this paper centers on those entries which are relevant to MODFs. The H-K Schema follows the Attribute Convention for Data Discovery (ACDD; Earth Science Information Partners (ESIP), 2022), the NetCDF Climate and Forecast (CF) Metadata Conventions (Eaton et al., 2022), and the ISO/TC 211 19100 series of standards for digital geographic information. Short variable names are compliant with the World Climate Research Program (WCRP) Coupled Model Intercomparison Project, Phase 6 (CMIP6; Taylor et al., 2022) whenever possible as these are in wide use in the geophysical research community. For global attributes, the H-K Schema specifies identifying the file feature type, the file maker, the license, the location, information on when data was collected, and a version number associated with the MODF and its permanent identifier allowing for MODF augmentation. For variable attributes, the H-K Schema specifies identifying vocabularies, units, individual variable attribution (who originally collected the data) and variable provenance, and also presents a method for differentiation of multiple measurements of the same geophysical variable (redundancy). The H-K Schema is available in both json (machine readable) and pdf (human readable) formats. The A-M Variable and



Attribute Template Table (Morris and Akish, 2022) was developed iteratively in conjunction with the H-K Schema for
collating metadata and was used as a direct input to the MODF creation process.

**5.1 Global Attributes**

Global attributes are the descriptive metadata that are relevant for the entire MODF file. Table 1 lists recommended global
attributes for MODFs.  These were chosen because they are highly recommended by ACDD; because they are recommended
by ACDD and made sense for our purpose, and seemed not overly burdensome for those most likely to be making MODFs;
and/or because we felt they would help MODFs be FAIR.  Some global attributes whose inclusion and consistent use is
particularly important are described below. We use italics for the names of attributes and single quotes around variable
names.

*id*: The global unique Persistent Identifier (PID).  This global attribute does not supersede references to the PIDs associated
with individual variables (Prakesh, et al., 2016), which can be provided within variable attribute *references*.  The MODF *id*
remains constant as modifications or additions are made to the MODF file.

*license:* Specifies terms of distribution.  As with *id,* individual variables and groups of variables in the MODF may have
unique license requirements that can be recorded in the variable attribute *comment*.


*creator_name:* The names of individuals who should be in the citation for the MODF (c.f. Jones et al. 2018, §2.2.7.3.3).  The
DataCite Metadata Working Group (2021) defines creators in this context as the main producers of the MODF file and also
links them to authorship.  (Note that the main producers of component data sets are credited within the variable attributes
coming from their individual data sets.)  Enabling those who create MODFs to receive appropriate credit for their work is a
key element of making MODFs FAIR; data reuse depends in part on data accessibility, which in turn is more likely when
data are "considered legitimate, citable products of research" (Data Citation Synthesis Group, 2014).

*featureType:* Although the original vision was to have a single MODF for a site during a campaign that would include all
relevant variables, when considering the practicalities of archive submission, it became apparent that file *featureType* needed
to be specified to facilitate data services such as visualization tools.  Most observatory data fits into *featureTypes* timeSeries,
timeSeriesProfile and timeSeriesTrajectory. These *featureTypes* are defined by the temporal-spatial dimensions that are
associated with individual variables: time only for timeSeries, time and height (or depth) for timeSeriesProfile, and time–
height (or depth)–latitude–longitude for timeSeriesTrajectory.

*missing_value:* Establishes one consistent *missing_value* for all data in the MODF.



**Table 1: H-K Schema Recommended Global Attributes (adapted from Hartten and Khalsa, 2022)**

| Attribute | Description of Required Information |
|---|---|
| *title* | Short human-readable phrase or sentence describing the MODF |
| *date_created* | Date on which current version of the MODF was created or modified |
| *Conventions* | List of the conventions that are followed by the MODF |
| *standard_name_vocabulary* | Name and version of the controlled vocabulary from which variable Standard Names come |
| *creator_name* | Name of the main individual(s) involved in producing the MODF, or the authors of the publication, in priority order |
| *creator_email* | Email of the main individual(s) involved in producing the MODF, or the authors of the publication (if there is one) describing the MODF |
| *institution* | Institution where the original MODF was produced |
| *id* | Persistent Identifier (PID) for the MODF |
| *naming_authority* | Naming authority for PID (preferably using reverse-DNS naming, e.g. 'edu.ucar.unidata', although URIs may be used) |
| *license* | Terms of distribution and use |
| *time_coverage_start* | Time of the first data point in the MODF |
| *time_coverage_end* | Time of the last data point in the MODF |
| *featureType* | One of the following: 'point', 'timeSeries', 'trajectory', 'profile', 'timeSeriesProfile', or 'trajectoryProfile' |
| *contributor_name* | Name of any individual(s) responsible for collecting, managing, distributing, or otherwise contributing to the development of the MODF; may indicate those who helped with the development but who were not so "key" as to be listed as an author |
| *contributor_email* | Email of any individual(s) responsible for collecting, managing, distributing, or otherwise contributing to the development of the MODF |
| *project* | Name of the project(s) principally responsible for originating this MODF |
| *summary* | A paragraph describing the MODF |
| *source* | The method of production of the MODF |
| *metadata_link* | URL to detailed documentation (DOIs expressed as http://doi.org/...) |
| *history* | Provides an audit trail for modifications to the MODF |
| *references* | Published or web-based references that describe the MODF or methods used to produce it |
| *keywords* | Comma-separated list of key words and/or phrases, preferably drawn from a controlled vocabulary, e.g. the GCMD at https://earthdata.nasa.gov/earth-observation-data/find-data/gcmd/gcmd-keywords |



## 5.2 Time, Space and Site Variables

In the H-K Schema, variables have been divided into subgroups. There are three subcategories with time, space and site information: temporal dimensions and variables; spatial dimensions and variables; single level fixed variables. Per the
NetCDF Users Guide (Unidata, 2023), dimensions "may be used to represent a real physical dimension, for example, time, latitude, longitude, or height.... [or] to index other quantities, for example station or model-run-number." Examples of these indexing variables are listed in Table 2 and discussed further below. Per the H-K Schema, all dimensional and geophysical variables are given a short CMIP or CMIP-like variable name and are characterized by variable attributes *long_name*, *standard_name*, *units*, additional recommended attributes, and any other attributes the MODF creators feel necessary to fully
document the data.

**Table 2: Examples of dimensions and variables related to time, space, and site information, together with the associated *long_name*, *standard_name*, *units* and recommended attributes (extracted from Hartten and Khalsa, 2022).**

| Variable Name (CMIP or CMIP-like) | *long_name* Attribute | *standard_name* Attribute | *units* Attribute | Minimum Recommended Additional Attributes |
|---|---|---|---|---|
| TEMPORAL DIMENSIONS AND VARIABLES | | | | |
| time | Valid Time | time | hours since ... | |
| time15 | Valid Time for Observations with 15-Minute Cadence | time | hours since ... | *delta_t*; *calendar* |
| time_sonde | Radiosonde Valid Time | time | hours since ... | |
| SPATIAL DIMENSIONS AND VARIABLES | | | | |
| height_tower | Tower Observation Height | height | m | _sonde variables only: |
| lat_sonde | Radiosonde Latitude | latitude | degrees_north | *missing_value*; |
| lon_sonde | Radiosonde Longitude | longitude | degrees_east | *actual_range*; *instrument*; |
| alt_sonde | Radiosonde Altitude | altitude | m | *source* |
| SINGLE-LEVEL FIXED VARIABLES | | | | |
| orog | Surface Altitude | surface_altitude | m | *missing_value*; *source*; *references*; *comment* |



'time', 'time15', 'time_sonde': MODFs are intended to retain as much high-resolution information from the original data collection as possible. Averaging is limited to that necessary for processing (e.g. for eddy-correlation flux calculations) and data are not interpolated. Therefore, temporal dimensions support maintaining original data collection cadences for individual variables. For instance, in MODFs 'time' is a generic temporal dimension whereas 'time15' is a time dimension associated with variables collected at 15 min intervals and 'time_sonde' is the time dimension associated with the data

collected during a radiosonde ascent. In the YOPPsiteMIP implementation of MODFs, we have chosen to append 'N' or '_platform' to a generic CMIP name such as 'time' in order to indicate a time array with a particular cadence in minutes or a time array tied to a particular instrument or platform. In keeping with CF Metadata guidelines, all the differently named time variables can and should have the same long_name attribute.

'height_tower': A spatial coordinate can be a single scalar value or a set of values (a dimension) that describes the location where geophysical measurements are collected. This can be an array of fixed values in the case when measurements are made at set levels, for instance on an instrumented tower. A tower may also consist of multiple measurements of the same geophysical variable collected at different heights above the ground. For instance, observations of air temperature collected by three sensors located at the surface (2 m AGL), on a pole (10 m AGL), and on top of a building (20 m AGL) could be

combined into a single array designated as '_tower' so long as the sensors' horizontal positions were co-located."

'lat_sonde': Spatial coordinates can also be a dimension variable or a scalar value. For instance, if information is retained on the latitude, longitude and/or altitude of a sonde during its ascent, this information should be put into coordinate variables that provide the location in space for the geophysical variables collected by the sonde. If the sonde output includes only

varying altitudes and a fixed launch time, the geophysical data from each sonde flight should be put into a file with the *featureType* "profile". If the sonde output includes varying altitudes and associated times, the geophysical data should be put into a file with the *featureType* "timeSeriesProfile". In both cases a scalar altitude variable should be used as the vertical dimension for each flight. However, if varying latitude and longitude are also provided, the geophysical data from each sonde should be put into a file with the *featureType* "timeSeriesTrajectory" with the reported altitude, latitude and longitude

being provided as coordinate variables for each flight.

'orog': Certain fixed variables are descriptive of the site or the platform from which measurements are taken. An example of the former is the variable with the surface altitude (CMIP name 'orog'), which describes the site's altitude above the surface defined as the lower boundary of the atmosphere. Other common single-level fixed variables are the 'lat' (latitude) and the

'lon' (longitude) chosen to generally identify a site location. If the site has distributed measurements, individual variables may have more refined 'lat_platform' and 'lon_platform' variables or dimensions.





### 5.3 Geophysical Variable Attributes and Examples

There are six categories of geophysical variables:  Single-level atmosphere variables; surface and Top-of-Atmosphere (TOA[3]) variables; atmospheric variables on model or instrument levels; sub-surface terrestrial variables; oceanic variables on

model or instrument levels; sea ice variables. Examples of *long_name*, *standard_name*, and *units* attributes for observed geophysical variables that have been catalogued in the H-K Schema are shown in Tables 3 and 4. A full listing of all the single-level atmospheric variables, the surface and TOA variables, and the atmospheric variables on instrument levels in the H-K Schema version 1.2 is presented in Table A1, while a listing of the current H-K Schema oceanic variables on model or instrument levels, sub-surface terrestrial variables, oceanic single-level variables, and sea ice variables is found in Table A2.

Some discussion of the examples in Tables 3 and 4 follows.

CMIP name: The Coupled Model Intercomparison Project (CMIP) names are taken from the CMIP6 Participation Guidance for Modelers, a program supported by the NASA Program for Climate Model Diagnosis & Intercomparison. CMIP names prioritize terse, presumably code-efficient abbreviations such as 'ta' (air temperature). To clearly identify the variables that

were observed or derived redundantly from different methods or platforms, a suffix can be appended (e.g., _tower, _radar, _8m).  Some standard measurements, such as 2m temperature and 10m winds, have unique CMIP names such as 'tas', 'uas' and 'vas' (near-surface air temperature, eastward wind, and northward wind, respectively). In the case where a CMIP variable name has not been defined in the CMIP6 vocabulary, a CMIP-like name has been composed.

*long_name:* Fully describes the physical quantity and can be thought of as useful attribute for labeling plot axis; in other words, it is the name that best communicates with humans about what the variable is (note that the *long_name* does not necessarily need to be in English). The H-K Schema provides ad hoc *long_name* definitions for all variables.

*standard_name:* Taken from the CF Standard Name Table (2023), which is periodically updated based on community

requests and discussion. Standard names are constructed in conformance with the CF Metadata Conventions (Hassell et al., 2017). Different variables can have the same standard name.  For instance, 'albs' and 'albsn' both have the *standard_name* of surface_albedo, but 'albsn' is restricted to use over snow-covered areas. Redundant variables (such as multiple measurements of temperature, or fluxes computed by bulk versus eddy-correlation methods) will have the same standard name and will require differentiation by a suffix added to the CMIP name, additional descriptors in the *long_name*, and

possibly accompanying spatial-temporal indices.

---

[3] TOA variables are only relevant to MMDFs, but are grouped with surface variables in the H-K Schema and therefore included for completeness.



**Table 3: Examples of atmospheric variables, together with the associated *long_name*, *standard_name*, and *units* attributes (extracted from Hartten and Khalsa, 2022).**

| Variable Name (CMIP or CMIP-like) | *long_name* Attribute | *standard_name* Attribute | *units* Attribute |
|---|---|---|---|
| SINGLE-LEVEL ATMOSPHERIC VARIABLES | | | |
| tas | Near-Surface (2m) Air Temperature | air_temperature | K |
| tas_site1 | Near-Surface (2m) Air Temperature at Site1 | air_temperature | K |
| uas | Near-Surface (10m) Eastward Wind | eastward_wind | m s-1 |
| SURFACE AND TOA VARIABLES | | | |
| snd | Surface Snow Thickness | surface_snow_thickness | K |
| ts | Surface (Skin) Temperature where Land or Sea Ice | surface_temperature | K |
| rsds | Downward Shortwave Radiation at the Surface | surface_downwelling_ shortwave_flux_in_air | W m-2 |
| ATMOSPHERIC VARIABLES ON MODEL OR INSTRUMENT LEVELS | | | |
| ta | Temperature | air_temperature | K |
| rsd | Downward Shortwave Radiation | downwelling_ shortwave_flux_in_air | W m-2 |
| ua | Eastward Wind Component | eastward_wind | m s-1 |





**Table 4: Examples of non-atmospheric variables, together with the associated *long_name*, *standard_name*, and *units* attributes (extracted from Hartten and Khalsa, 2022).**

| Variable Name (CMIP or CMIP-like) | *long_name* Attribute | *standard_name* Attribute | *units* Attribute |
|---|---|---|---|
| SUB-SURFACE TERRESTRIAL VARIABLES | | | |
| gtsl | Bulk Soil Temperature | soil_temperature | K |
| mrlsl | Layer-Average Soil Moisture | moisture_content_of_soil_layer | kg m-2 |
| OCEANIC SINGLE-LEVEL VARIABLES | | | |
| tos | Sea Surface Temperature | sea_surface_temperature | K |
| mlotst | Ocean Mixed-Layer Depth | ocean_mixed_layer_ thickness | m |
| rsntds | Net Downward Shortwave Radiation at Sea Water Surface | net_downward_ shortwave_flux_ at_sea_water_surface | W m-2 |
| OCEANIC VARIABLES ON MODEL OR INSTRUMENT LEVEL | | | |
| to | Ocean Temperature | sea_water_temperature | K |
| so | Sea Water Salinity | sea_water_salinity | 1e-3 |
| SEA ICE VARIABLES | | | |
| sithick | Sea Ice Thickness | sea_ice_thickness | m |
| sisali | Sea Ice Salinity | sea_ice_salinity | 1e-3 |

Table 5 lists the other attributes that should be included in MODFs for geophysical variables. Some of the variable attributes in Table 5 are listed in the CF Conventions as being for use as either global or data attributes, but the *history* attribute is

listed as for global use only. We have encouraged its use with variables (data) in MODFs because we believe that maintains the spirit if not the letter of the CF Conventions; like *institution*, *references*, *source*, and *title*, *history* helps document the provenance and nature of the data included in these multi-institution, multi-sourced files. We also encourage MODF (and MMDF) makers to make use of additional data attributes to share information about the variable contents with users.

*original_name:* Refers to the name of the variable in the original file from which it was extracted. This provides an important cross reference in the case where a user may need to refer back to the original source data set.



*instrument:* Tracking instrumentation characteristics and in many cases calibration coefficients is critical to provide information for users who may be in the process of developing refined data processing, developing higher-order products, or doing instrument intercomparison studies.

*references:* Published material that describes either the data or the methods used to produce it should be listed here. Best practice is to include a URI (a DOI for a paper, or a URL for a website).

*source:* Both the CF Conventions and ACDD describe this as "the method of production of the original data", by which they mean either the model that produced it or the instrument that gathered it. In either case, the idea is to give the user information that will help them understand what they're working with, what assumptions or methods are inherent to the data. Therefore, the name and version of a numerical model, or the type and perhaps make and model of an instrument, would be appropriate here.

*history:* Provides an audit trail for the data, documenting its provenance. When used as a variable attribute, this should trace what has been done to the data from its raw state until it was put into this file. Ideally, each step should be documented by a line containing a datestamp, the person or entity who did the action, a brief description of the action, and any program(s) which accomplished the action (including relevant settings or command arguments). When used as a global attribute, this is the attribute in which changes to the file should be recorded.

**Table 5: Minimum recommended additional variable attributes for geophysical variables (adapted from Hartten and Khalsa, 2022)**

| Additional Variable Attributes | |
| --- | --- |
| *missing_value* | *contributor_name* |
| *actual_range* | *contributor_email* |
| *instrument* | *creator_name* |
| *source* | *creator_email* |
| *references* | *institution* |
| *history* | *comment* |
| *original_name* | |

Figure 3 describes the work flow and process that developed through the efforts of the YOPPsiteMIP team of observers, modellers, data scientists, and data managers. This is a highly iterative process; note the two-way arrows between the H-K Schema and both the A-M Table and the MODF Checker.





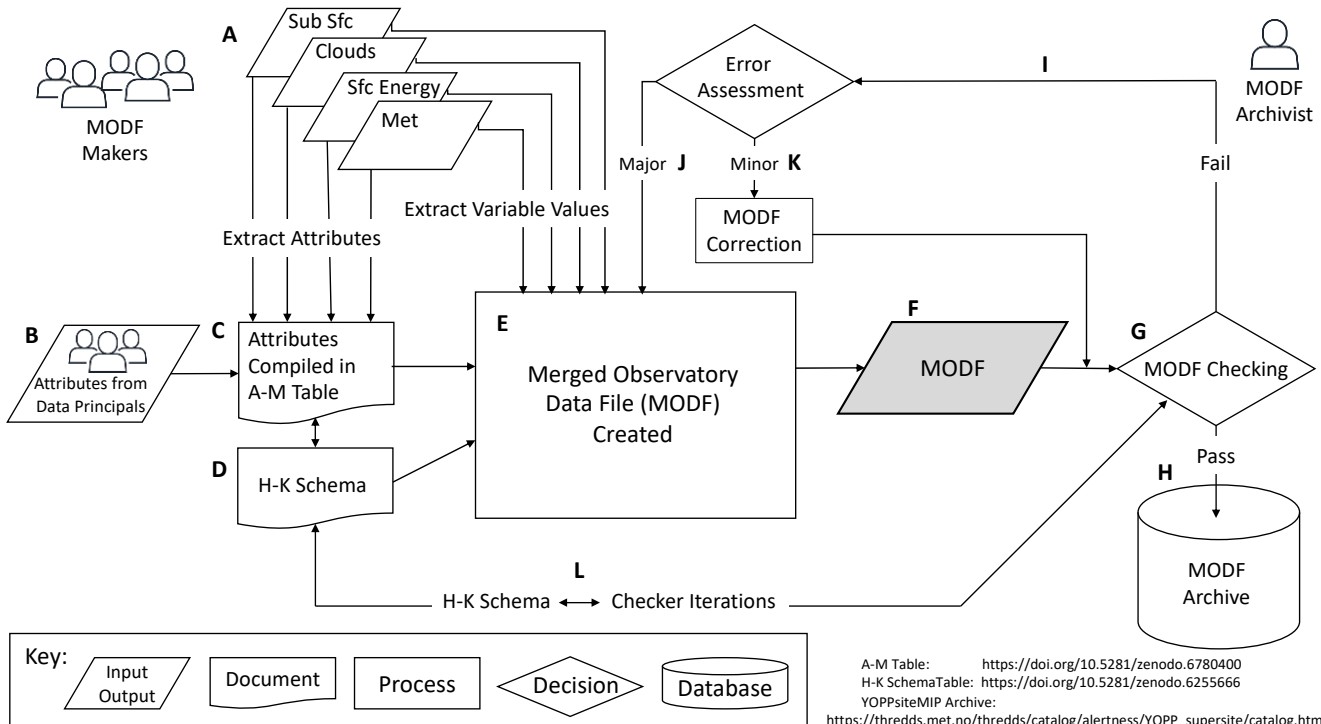

**Figure 3: The MODF workflow process. A: Gather input files, B: Interview contributing data principals for necessary metadata not digitally encoded in data files, C: Utilize the A-M MODF template to construct the specific MODF framework based inputs from A and B compliant with the H-K Schema (D). MODF Makers create the MODF (E) with inputs from (C) and the data values from individual files collected (A). F: MODF file. G: MODF checking. H: Upon 'pass', send MODF to archive. I: Upon 'fail' assess MODF. J: Major fails, return MODF Makers. K: Minor fails, archivist corrects and sends MODF to archive after rechecking (G). L: Iterative developments between H-K schema (D) and MODF checking (L). People icons adapted from one designed by vectorstock (Image #45970079 at VectorStock.com).**

## 6 Discussion

We have presented the H-K Schema (Hartten and Khalsa, 2022) and a production framework for organizing complex campaign and observatory data from multiple instruments into Merged Observatory Data Files (MODFs). The H-K Schema can also be used to format forecast model output into corresponding Merged Model Data Files (MMDFs). MODFs and MMDFs are compliant with existing metadata and data standards that support FAIR principles. The Schema and the framework were developed by a YOPPsiteMIP working group of observers, modelers and data managers. MODFs address the mundane but complex issues that arose from the YOPPsiteMIP vision to confront polar weather forecasting models with observations from richly instrumented sites during special observing periods. The issues addressed include semantics, attribution for original data, data provenance, different cadences, multiple measurements of the same variable from the same site (including local, subgrid-scale spatial distribution), versioning strategies to account for different levels of data



processing, unit conventions, and missing data indicators. Because of the high cadence of many measurements (seconds to minutes), MODFs can be used to evaluate accumulating biases at sub-model timestep increments.

Although providing site, instrument, processing, and attribution metadata documentation is good practice, it is often neglected. Since this essential MODF information is difficult to assemble after the fact, we recommend that observers use datagrams during development, deployment, and operation of sensors as a standard practice. Datagrams "are designed to document the life story of a data value from start to finish and provide a guide for humans to design, deploy, troubleshoot, repair, record, transmit, process, and archive data collected with measuring devices." (Morris and Uttal, 2022)


Currently the H-K Schema includes geophysical variables across the atmosphere (state, clouds, turbulence, radiation), snow, ice, terrestrial, and ocean systems. Appendix A lists the MODF (and MMDF) values that are currently in the H-K Schema (version 1.2). These are not exhaustive, and we expect that MODF users and makers will augment their MODF files to accommodate individual campaigns by including, for example, atmospheric aerosols, constituent gases, additional ocean and

terrestrial variables, and ecosystem and biogeochemical data. Additional variables should be incorporated with the metadata standards described here.

A first set of MODF and MMDF files implementing the H-K Schema from concept to production for YOPP Special Observing Periods have been archived by the Norwegian Meteorological Service where a collection[4] of matched MODF and

MMDF files are available for several Arctic ground stations. An initial analysis (Day et al., submitted) using the YOPPsiteMIP MODF–MMDF collection demonstrates through a number of examples and case studies how process-describing variables such as surface fluxes (longwave, sensible and latent) and ground fluxes can be used to gain deeper understanding into the nature of forecast errors. Although we strongly encouraged YOPPsiteMIP MODF and MMDF Makers to check their files for compliancy before submitting, we have found it expedient to have the recipient data managers

at the host archive take the final responsibility for inspecting, certifying, and in some cases making editorial corrections to bring the MODFs and MMDFs into compliance with the H-K Schema. Unfortunately when archives have rigorous standards for submittal, data providers often do not submit at all because they do not have the resources to support that level of data management.

Generating MODFs will expand usage of data from field campaigns by increasing data uptake and decreasing data latency. This will promote usage of non-operational data that are currently underutilized and difficult to access comprehensively, specifically for environmental services requiring near real-time environmental intelligence. The U.S. Interagency Arctic Research Policy Committee (IARPC) has defined environmental intelligence as "a system through which information about

---

[4] https://thredds.met.no/thredds/catalog/alertness/YOPP_supersite/catalog.html





a particular region or process is collected for the benefit of decision makers through the use of more than one inter-related

source". Furthermore, IARPC notes that "Traditionally, researchers collect data, develop models, and communicate results through well-established channels that are often slow and inefficient. While the vetting of scientific results ensures that the conclusions are of highest quality, the process is not well-aligned with the need for rapid information delivery in the face of environmental transitions that are putting stress on ecosystems and human populations." MODFs will not only accelerate timely and relevant data access and scientific results for the primary researchers that collected the data, but will also support

the iterative observations, modeling and data systems that connect researchers, stakeholders and decision-makers to allow informed responses to environmental events. To this end, MODFs are designed to be living files that can be created in a timely manner with near real-time variables and then augmented when additional variables become available. This is particularly important for situations in which observational data becomes obsolete before it can be utilized by environmental awareness and short-term prediction services for extreme events. MODFs can then be augmented with variables that are only

available after extensive human-assisted processing (e.g., surface energy balance fluxes) for research purposes, as well as with new versions of variables that require detailed quality control to produce higher level and higher reliability products. This addresses the concern that many data providers express about only releasing the most highly curated values (often resulting in data embargos and release time lags) and recognizes that the level of necessary post processing depends on the application.


We expect that MODFs and MMDFs, in other words Merged Data File (MDF) collections, will also be used to facilitate the concept of system science. The NSF workshop report Opportunities and Challenges of Arctic System Science (Vorosmarty et al., 2018) introduced the concept of "… multiple 'currencies' that link the Arctic climate and environment—geophysical entities such as water, energy, carbon, and nutrients with quantifiable properties—and how they interact to produce and

illuminate systems-level behaviors". MODFs, by quantifying the currencies throughout a system with internally consistent standards, can serve to break down barriers between individual studies of separate components of the system that are typically divided along disciplinary lines, thereby advancing multidisciplinary process studies.

The concept of organizing multivariate observational data sets from a single site or platform into a unified product is not

unique. Wei et al. (2021) describes how ICARTT data format conventions (Aknan et al., 2013) can be used to create merged in-situ data products from research aircraft carrying multiple sensors. The U.S. DOE ARM program (Stokes and Schwartz, 1994) creates Climate Modeling Best Estimate (CMBE) data files (Xie et al., 2010) for their Climate Reference Sites. Hogan and O'Connor (2004) describe how the Cloudnet project preprocesses cloud data from multiple remote sensors using appropriate ancillary observations and model forecasts, then makes the output available in netCDF format with rich metadata

about the preprocessing. What is unique about MODFs is the strategy of creating corresponding matched MMDF data files to support modeling verification and process evaluations. This addresses a long-standing issue with communication silos between observing and modeling sciences (Holloway et al., 2014, Sprintall et al., 2021, Neang et al., 2021).



**Appendix A: Current Measurable Geophysical MODF and MMDF Variables and Their "Essential Climate Variable" Status**

The H-K Schema is a set of tables that is expected to be expanded and adjusted as the need arises for additional MODFs and MMDFs and for additions to existing ones. Therefore, researchers interested in using the H-K Schema to guide file creation should always refer to the current online version; the DOI listed in the References of this article will always land on the most recent vesion of the Schema, and when a new version of the H-K Schema is published the earlier versions will be prefaced with a note including a link the the latest version.


In this Appendix we list the *long_name* attribute for all atmospheric (Table A1) and non-atmospheric (Table A2) geophysical variables, current as of the time of this manuscript's submission. We do this for two reasons: to show the variety of variables incorporated in the H-K Schema so far, and also to highlight the variables in the Schema which the WMO has identified as Essential Climate Variables (ECVs; Essential Climate Variables, 2023). Note that some of the ECVs are

defined by the WMO in a manner other than the point measurements typically made at field sites. Those details are identified by footnotes in the tables. Lavergne et al. (2022) have proposed expanding the list of ECVs related to sea ice; proposed sea ice ECVs that are in the H-K Schema are separately highlighted in Table A2.






**Table A1: Atmospheric variables (extracted from Hartten and Khalsa, 2022) that are measured at Arctic YOPP supersites or included in Arctic YOPP MMDFs (prefaced by "•"). Underlined variables are ECVs.**

| Variables' *long_name* Attribute | |
|---|---|
| SINGLE-LEVEL ATMOSPHERIC VARIABLES | |
| Surface pressure | Height of atmospheric boundary layer |
| Mean sea level pressure | Total precipitation of water in all phases per unit area |
| Near-surface eastward wind | Total cloud cover |
| Near-surface northward wind | Cloud optical thickness |
| Near-surface wind speed | Total column water vapour |
| Direction near-surface wind from | Total column cloud water in liquid phase |
| Near-surface wind gust | Total column icewater |
| Near-surface air temperature | Surface horizontal visibility |
| Near-surface dew point temperature | Ozone concentration in air[a] |
| Near-surface specific humidity | Photosynthetic photon flux density |
| Near-surface relative humidity | Reflected photosynthetic photon flux density |
| Surface roughness for momentum | |
| Surface roughness for heat | |
| SURFACE AND TOP-OF-ATMOSPHERE VARIABLES | |
| Surface snow thickness | •Top-of-atmosphere incoming shortwave radiation |
| •Surface snow area fraction | •Top-of-atmosphere outgoing shortwave radiation |
| Snow water equivalent | •Top-of-atmosphere outgoing long wave radiation |
| Snow density | Upward surface shortwave radiation |
| Surface (skin) temperature where land or sea ice | Downward shortwave radiation at the surface |
| Snow surface skin temperature | Net shortwave radiation at the surface |
| Snow temperature | Upward surface longwave radiation |
| Ground skin temperature | Downward surface longwave radiation |
| Surface albedo[b] | Net longwave radiation at the surface |
| Snow albedo[b] | •Surface turbulent latent heat flux |
| Surface downward heat flux in snow | Surface turbulent latent heat flux[c] (bulk method) |
| Downward heat flux at snow botton | Surface turbulent latent heat flux (eddy covariance method) |
| •Canopy area fraction | •Surface turbulent sensible heat flux |
| Time-average eastward turbulent surface stress | Surface turbulent sensible heat flux (bulk method) |
| Time-average northward turbulent surface stress | Surface turbulent sensible heat flux |
| | (eddy covariance method) |
| | Ground heat flux |

[a] Total column ozone is the listed ECV, while observations from a site are typically point values.

[b] The ECV listing is for maps, not point measurements, of albedo; snow albedo is not explicitly listed.

[c] The ECV listing specifies "Land-Biosphere Evaporation from Land".



---

### Variables' *long_name* Attribute

#### ATMOSPHERIC VARIABLES ON MODEL OR INSTRUMENT LEVELS

- •Geopotential height
- Geopotential height on half levels
- Atmospheric pressure
- Pressure on full levels
- Pressure on half levels

- Eastward wind component

- Northward wind component

- Wind speed
- Direction wind from
- Vertical velocity
- •Vertical large-scale wind in pressure coordinates
- Air temperature

- Dew-point temperature
- Specific humidity
- Relative humidity
- Wet-bulb potential temperature
- •Tendency of air temperature
- •Tendency of air temperature due to advection
- •Tendency of specific humidity
- •Tendency of specific humidity due to advection
- •Tendency of eastward wind
- •Tendency of northward wind

- Upward longwave radiation
- Downward longwave radiation
- Upward shortwave radiation
- Downward shortwave radiation
- •Vertical eddy diffusivity coefficient for momentum due to parameterized turbulence
- •Vertical eddy diffusion coefficient for temperature due to parameterized turbulence
- Turbulent sensible heat flux based on virtual potential temperature
- Turbulent moisture flux based on vapor content
- Eastward turbulent momentum flux
- Northward turbulent momentum flux
- Turbulent kinetic energy
- •Percentage cloud cover, including both large-scale and convective cloud
- Mass fraction of cloud liquid water
- Mass fraction of cloud ice
- Snowfall flux per unit area
- Cloud base height

---





**Table A2: Non-atmospheric variables (extracted from Hartten and Khalsa, 2022) that are measured at Arctic YOPP supersites or included in Arctic YOPP MMDFs (prefaced by "•"). Underlined variables are ECVs, with a dashed underline used for newly proposed sea ice ECVs (Lavergne et al., 2022).**

| Variables' *long_name* Attribute | |
| --- | --- |
| **SUB-SURFACE TERRESTRIAL VARIABLES** | |
| Temperature of soil | Average layer soil moisture |
| **OCEAN SINGLE LEVEL VARIABLES** | |
| Sea surface temperature | Atmosphere-ocean sensible heat flux |
| Ocean mixed-layer depth | Atmosphere-ocean latent heat flux |
| Ocean surface x-stress | Net downward shortwave radiation at sea water surface |
| Ocean surface y-stress | Net downward longwave radiation at sea water surface |
| Significant wave height[d] | Fresh water flux into sea water |
| | •Water flux into sea water due to sea ice thermodynamics |
| **OCEAN VARIABLES ON MODEL OR INSTRUMENT LEVELS** | |
| Ocean temperature | Ocean u-velocity |
| Sea water salinity | Ocean v-velocity |
| | Ocean w-velocity |
| **SEA ICE VARIABLES, REPORTED ON ATMOSPHERIC GRID** | |
| Sea ice concentration (area fraction) | Rainfall rate over sea ice |
| •Sea ice concentration (area fraction) in categories | Sea ice surface temperature (at the interface of sea ice or the snow on it and the overlying air) |
| Sea ice thickness | Temperature at snow-ice interface |
| •Sea ice thickness in thickness categories | Temperature at ice-ocean interface |
| Snow thickness on sea ice | Sea ice/snow albedo |
| Sea ice age | Ocean-ice net sensible heat flux |
| Sea ice u-velocity | Net upward sensible heat flux over sea ice |
| Sea ice v-velocity | Net upward latent heat flux over sea ice |
| Sea ice salinity | Downwelling shortwave flux over sea ice |
| •Sea ice normal stress (pressure) | Upwelling shortwave flux over sea ice |
| •Compressive sea ice strength | Downwelling longwave flux over sea ice |
| Fast ice concentration (area fraction) | Upwelling longwave flux over sea ice |
| Fast ice thickness | Net conductive heat flux in ice at the surface |

---

[d] ECV listing is for just wave height.



**Author Contributions**

TU conceptualized the original MODF vision; CJC and MM participated in conceptualization by providing valuable perspectives on the practicality, utility and application of the MODF concept. BC, GS and JD provided methodology requirements from a numerical modeling perspective. LMH and SJK, in developing the H-K Schema, provided methodology requirements from an observational perspective and also enhanced data curation. JT and OG guided development of the H-K Schema and MODF formats by providing methodology requirements from a data repository standpoint.

Software contributions were provided by JH and JT, who wrote python code functions to facilitate MODF creation, checking and modification. Data curation, investigation, and software contributions were provided by EA, SM, NH, LXH, RC, ZM, EO, RP, JH and MM, who served as MODF Makers and through that practical application of the MODF concept developed the workflow indicated in Figure 3. They also iteratively worked with LMH and SJK to further develop the H-K Schema. JT contributed to data curation, software, and validation by serving as the MODF Curator and by iteratively working with LMH and SJK to further develop the H-K Schema. GP and NH enhanced data curation by developing strategies to have MODFs include comprehensive formal attribution for individuals and institutions.

TU drafted the original manuscript, and TU, LMH, and SJK revised the draft, with commentary and revisions from BC, GS, JD, SM, EO, RP, ZM, JT, MM, and CJC. Visualizations, in the form of figures and tables, were drafted by TU and revised by TU and LMH.

**Competing interests**

The authors declare that they have no conflict of interest.

**Code and data availability**

The H-K Variable Schema Table developed for the YOPPsiteMIP is archived with Zenodo:
https://zenodo.org/records/6463464
The Table is licensed under the Creative Commons Attribution 4.0 International License

A preliminary set of MODF and MMDF files developed for the YOPPsiteMIP is available at:
https://thredds.met.no/thredds/catalog/alertness/YOPP_supersite/catalog.html



The catalog link provides information on licensing and crediting which is not repeated here as the example files are not the subject of this manuscript. Acknowledgements

**Acknowledgements**

This is a contribution to the Year of Polar Prediction (YOPP), a flagship activity of the Polar Prediction Project (PPP), initiated by the World Weather Research Programme (WWRP) of the World Meteorological Organization (WMO).

This work was supported in part by NOAA's Global Ocean Monitoring and Observing Program (FundRef https://doi./org/10.13039/100018302), the NOAA Physical Sciences Laboratory (TU, LMH, EA, SM, CJC), and

the NOAA Global Monitoring Laboratory (SM). LMH, EA, and SM were supported in part by NOAA cooperative agreements NA17OAR4320101 and NA22OAR4320151; LMH was also supported by NOAA's Climate Program Office, Climate Observations and Monitoring Program (FundRef 100007298). This work was also supported in part by the U.S. Department of Energy's Atmospheric System Research, an Office of Science Biological and Environmental Research program. JD was supported by European Union's Horizon 2020 Research and Innovation program through Grant

Agreement 871120 (INTERACTIII). RP was partly supported by the European Union's Horizon 2020 Research and Innovation program projects INTAROS (grant 727890) and PolarRES (grant 101003590).

Michael Gallagher (Univ. of Colorado/CIRES and NOAA Physical Sciences Laboratory) established the MODF Makers GitLab and Slack channels and wrote python functions to facilitate MODF creation. We are grateful to Dave Allured (Univ.

of Colorado/CIRES and NOAA Physical Sciences Laboratory) for extensive discussions about the CF Conventions, which greatly improved the MODF and MMDF projects. We also thank Scott Landolt (NCAR/Research Applications Lab) for discussions about measuring snow.

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
