# Peer review of "Merged Observatory Data Files (MODFs): An Integrated Observational Data Product Supporting Process-Oriented Investigations and Diagnostics"

_EGUsphere, 2023_

## Referee Comment (RC1)

Review of "Merged Observatory Data Files (MODFs): An Integrated Observational Data Product Supporting Process-Oriented Investigations and Diagnostics" by Taneil Uttal et al.

The MODF concept is laudable and should be supported. The paper as drafted does a good job of discussing the technical details of what an MODF file is. It does, however, skirt over or around a number of fairly hefty issues which it would be worth trying to address in redrafting in my view while, of course, trying not to detract from the main technical nature of the piece.

**Major comments**

1. The preparation of files in a highly usable format is a necessary but not a sufficient condition for the broad scale use of observatory and campaign data. While absolutely not the focus of the paper it feels very remiss not to have a brief 1-2 paragraph discussion around the broader aspects surrounding exchange, archival and dissemination of these data. Basically, if every single observatory and campaign retrospectively reformatted everything to the proposed format we would still not have solved the broader problems around discoverability and accessibility of these data which requires a systemic effort to collate and provide more unified access to the data. I would think a couple of paragraphs discussing next steps to enable exploitation using the MODF concept as a way to harmonise data formatting issues would strengthen rather than distract from the piece. You'd basically be making the case that MODF is an enabling step in a broader activity to enable greater use of these data by the community. This may include a dedicated effort to collate such data from multiple existing and planned observatories and campaigns and provide access via a single unified repository which may well be federated in a similar manner to CMIP itself.

2. While issues of cadence or reporting frequency are dealt with, it is unclear how broader collocation issues are dealt with in the proposed MODF file format. Take an example of measuring upper-air variables with a lidar, a radiosonde, an FTIR and a monumented GNSS sensor. While all may nominally sense water vapour the measured volumes as well as the time intervals differ substantively (balloons drift, lidars measure vertically, FTIR is in direction of sun. GNSS depends upon multiple complex path angles that are ever varying). It is not sufficiently clear how these distinctions are dealt with in a file or how a user is guided to account for the fact that there will be differences arising from what was measured rather than the measurements themselves. See Immler et al 2005 for further discussion in the context of metrological comparison closures in developing GRUAN products.

3. If MODF files enable version replacement for subcomponents then how are MODF files themselves proposed to be versioned and archived to enable reproducibility? If very actively curated there could be tens or even hundreds of unique versions of MODF files as different subcomponents are periodically reprocessed and reissued? The description is a little unclear to me as given how this will be handled. Maybe its covered in Section 4 but if so its not sufficiently clear to me as presently drafted and it would be beneficial to redraft for clarity.

4. Is the H-K schema a subset of GeoJSON or other emerging standards? It might be worth being a little more explicit. At least some of the names appear to be consistent with GeoJSON.
5. Despite the metadata retention being substantial it is not holistic. There are many metadata features not captured in the files as proposed which might be of use to researchers. Has thought been given to how to associate additional free-text / rich metadata with MODF files?

**Minor comments**

1. Line 28 – ECVs are defined by the Global Climate Observing System and not by the WMO
2. In line 326 what is the section reference in the parentheses to? Or is this a legacy needing removal? Its unclear to me.
3. In Table 2 the final column is completely screwed up with random row allocations that make no logical sense
4. In Table 2 discussion paragraph starting line 371 this should surely be 'lat_sonde' and 'lon_sonde'? It might also be worth nothing whether MODFs can cater for descent data which is increasingly being used and exploited.

---

## Author Response (AR1)

Prepared for inclusion with the final revision of egusphere-2023-2413:

**Merged Observatory Data Files (MODFs): An Integrated Observational Data Product Supporting Process-Oriented Investigations and Diagnostics**

Taneil Uttal et al.

Reviewer comments are in *italics*; our responses and the actions we took follow each comment. Line numbers refer to the "clean" copy of the revised manuscript, i.e. the one without change tracking.

*Reviewer #1:*
*Major comments:*
*1. Excerpt:* *While absolutely not the focus of the paper it feels very remiss not to have a brief 1-2 paragraph discussion around the broader aspects surrounding exchange, archival and dissemination of these data…. I would think a couple of paragraphs discussing next steps to enable exploitation using the MODF concept as a way to harmonise data formating issues would strengthen rather than distract from the piece.*

RESPONSE: We very much appreciate that Referee #1 recognizes the potential of a much wider scope of for usage of the MODF concept by data centers but these specifics of data management are out of the expertise (and influence) of the authors (field scientists and researchers using forecast center model data with field data to improve models).

ACTION TAKEN: We have added text to the Discussion suggesting that the data management community consider incorporating the MODF framework into data center operations. (See Section 6 "The main motivation …", lines 552-555, in revised manuscript)

*2. Excerpt:* *… it is unclear how broader collocation issues are dealt with in the proposed MODF file format…[While] a lidar, a radiosonde, an FTIR and a monumented GNSS sensor … all may nominally sense water vapour, the measured volumes as well as the time intervals differ substantively (balloons drift, lidars measure vertically, FTIR is in direction of sun. GNSS depends upon multiple complex path angles that are ever varying). It is not sufficiently clear how these distinctions are dealt with in a file or how a user is guided to account for the fact that there will be differences arising from what was measured rather than the measurements themselves.*

RESPONSE: The specific complexities of the multiple ways of measuring water vapour given by the reviewer as an example are real and important. However, the only way a user can be "guided to account for the fact that there will be differences [in the distinctions between the different observations of the same physical quantity] arising from what was measured rather than the measurements themselves" is to provide a structure in which the MODF makers can share information that will make clear to any committed user the differences between the platforms' measurement techniques, whether they be volumetric or temporal or spatial. Some user interest and engagement is required; an MODF maker cannot possibly extract all the possible "answers" every user might want about all the different variables and present it properly in the file. Instead, MODF makers need to point users towards the expert documentation likely to explain fine and complex details about the data. The H-K Schema already employs multiple ways of doing so by drawing from options in a few metadata standards, at least one of which is community-driven and open to user suggestions for additional metadata options.

ACTION TAKEN: We have addressed in the Discussion the fact that users may need to find or create new variable attributes to describe complex data, and pointed to explicit directions on how to do so via the CF Conventions, while at the same time re-iterating the

broad usefulness of the *references*, *source*, and *comment* variable attributes for sharing rich information. (See Section 6 "Currently the H-K Schema …", lines 533-539, and Section 5.3, lines 451-459 and 470-473, in revised manuscript)

**3. Excerpt:** *If MODF files enable version replacement for subcomponents then how are MODF files themselves proposed to be versioned and archived to enable reproducibility?…. The description is a little unclear to me as given how this will be handled. Maybe its covered in Section 4 but if so its not sufficiently clear to me as presently drafted and it would be beneficial to redraft for clarity.*

RESPONSE:  Versioning information is done primarily, but not exclusively, using the *history* attribute.

ACTION TAKEN:  We have reviewed the ACDD and the CF Convention material on attributes and made several additions and alterations to the section dealing with the H-K Schema.  The form of these changes is a bit different from what we proposed in our posted Author Comment, but this is because as we reviewed the material and considered both other changes inspired by reviewer comments and the overall structure of the manuscript, this seemed the clearest and least confusing way to improve reader understanding.  (See Section 5's second introductory paragraph, lines 308-312; new *history* paragraph in Section 5.1, lines 349-355; and revised *history* paragraph in Section 5.3, lines 476-480, in revised manuscript)

**4. Excerpt:** *Is the H-K schema a subset of GeoJSON or other emerging standards? It might be worth being a little more explicit.*

RESPONSE:  No, the H-K Schema is neither a subset of GeoJSON nor is it making use of GeoJSON, and all the standards it relies on are called out in the paper. The H-K Schema does not itself convey geographic information, thus GeoJSON adds nothing. And MODFs themselves are in netCDF, which doesn't use GeoJSON for encoding geographic information.

ACTION TAKEN:  None.

**5. Excerpt:** *There are many metadata features not captured in the files as proposed which might be of use to researchers. Has thought been given to how to associate additional free-text/rich metadata with MODF files?*

RESPONSE:  Free-text metadata can be added via the global attribute *comment* (not formally required by the H-K Table, but present in both ACDD and the CF Conventions) or via the variable attribute *comment* (listed in the H-K Table among the minimum required attributes for some variables).  Users can be directed to graphical or formatted textual metadata via the global attributes *references* or *metadata_link* (both recommended

in the H-K Table) or via the variable attributes *references* (also listed in the H-K Table among the minimum required attributes for some variables) or *comment*.

ACTION TAKEN:  We have added language clarifying that the requirements and recommendations of the H-K Table are minimums, and encouraging those wishing to make MODFs to explore the full range of options in the ACDD, the CF Conventions, and the DataCite Metadata Kernel.  (See Section 5's 2nd introductory paragraph, lines 308-310; the paragraph in Section 5.3 following Table 4, lines 452-458; and Section 6 "The main motivation …", lines 551-560, in revised manuscript)

*Minor comments:*
*1.  Line 28.*  *ECVs are defined by the Global Climate Observing System and not by the WMO*

RESPONSE:  Thank you.

ACTION TAKEN:  Done.  (See Abstract, line 3, and last paragraph of Appendix A, lines 627-629, in revised manuscript)

*2.  Line 326.*  *what is the seccton reference in the parentheses to? Or is this a legacy needing removal?*

RESPONSE:  That is to help direct the reader to the specific information in a large document.

ACTION TAKEN:  Changed to "(c.f. Jones et al. 2020, see §2.2.7.3.3)" to make the intent clearer.  (See Section 5.1, lines 332 in revised manuscript)

*3.  Table 2.*  *the final column is completely screwed up with random row allocations that make no logical sense*

RESPONSE:  Thank you for drawing attention to the fact that the final column was confusing to readers who hadn't been tweaking it for hours!

ACTION TAKEN:  A solid vertical gray line to the left of each list of additional recommended attributes has been used to indicate which rows that list applies to.  There have been some slight changes in the arrangement of items in the column and in the spacings of some rows associated with that change.  An explanatory line has been added to the table caption.  (See Section 5.2, line 356 in revised manuscript)

**4.  *Line 371+.*** *In Table 2 discussion paragraph this should surely be 'lat_sonde' and 'lon_sonde'? It might also be worth nothing whether MODFs can cater for descent data which is increasingly being used and exploited.*

RESPONSE:  These are both good points, thank you.

ACTION TAKEN:  The paragraph in question has been revised to explicitly include 'lon_sonde' and 'alt_sonde'.  We have also added a recommendation regarding how descent data from radiosondes should be dealt with, after first presenting a little more detail about the requirements of the CF Conventions. (See Section 5.2, lines 389-404, in revised manuscript).  In addition, we added a few sentences about the practice of saving descent data and the need for caution in using them earlier in the manuscript. (See Section 3.2, lines 181-185, in revised manuscript)

*Reviewer #2:*
*Specific comments:*
**1.  Line 60-64 (excerpt).**  *It could be highlighted further in other parts of the paper that a huge strength of MODF is that the "same variables from observations and models" be created and provided " in easy to use files of the same structure".*

RESPONSE:  Thank you for the suggestion.

ACTION TAKEN:  We have done this while also noting that it is important the models extract site data in real time during field experiments as it turns out to be very difficult to do after the fact.. (See Section 6 "We have presented …", lines 502-506; "Although providing …", lines 519-521; "The main motivation …", 551-553, in revised manuscript)

**2.  Line 90.**  *here and throughout, advocating for serial comma (a, b, and c): suggest inserting comma after rapid*

RESPONSE:  Thank you.

ACTION TAKEN:  Done, together with many other small grammatical and punctuation changes to improve correctness and clarity.  (See Section 2, lines 92, and throughout in revised manuscript)

**3.  Line 97.**  *The need for MODF might be even more important and necessary for coupled datasets, i.e. those dealing with more than 1 fluid and that attempt to also characterize fluxes and exchanges across the interface(s).*

RESPONSE:  Thank you for the suggestion.

ACTION TAKEN:  We believe that this is covered in what used to be the final paragraph of the Discussion, which has been very lightly edited and also moved up.  (See Section 6 "We also expect …", lines 580-586 in revised manuscript)

**4.  Line 136-163.**  *A challenge we face and have to address in our files is directionality, i.e. positive into ocean or atmosphere, what is negative or positive and why. This also requires standardization. It applies to both the radiative and turbulent fluxes.*

RESPONSE:  Directionality is addressed by the CF Conventions.  Most variables for which positive/negative matters (e.g. fluxes) already are pre-defined to indicate which is the positive direction.  The CF Standard Name Table clearly states that "The sign convention is that "upwelling" is positive upwards and "downwelling" is positive downwards" on all variables with up/down "-welling" or "-wards" in their names.  In addition, the text of the CF Conventions says that vertical coordinates other than pressure "must use the attribute **positive** which determines whether the direction of increasing

coordinate value is up or down". However, there's actually a certain amount of nuance ("legalese", if you will) in the places where coordinate definitions and the *positive* attribute are discussed, and it's best read in the original.

ACTION TAKEN: Added a sentences explaining how the CF Conventions address directionality in geophysical variables via the CF Standard Name Table, and how this information is also provided in the H-K Schema for emphasis and convenience. (See Section 5.3, *standard_name* paragraph, lines 439-443, in revised manuscript)

**5. Line 164-174.** *A challenge I see with precipitation datasets is ambiguity in units as a function of or including/contextualized in a time scale. Example: units of mm (rain accumulation) in an hourly dataset vs. a 1-min resolution mm/hr rain rate time series. These are different and units need to be specific.*

RESPONSE: This is an an issue a few of us have run into while working on incorporating many very specific types of precipitation variables into what will be the next version of the H-K Table! The issue is quite effectively and obviously addressed in the CF Conventions; precipitation variables come in both depth and rate "flavors", with clearly different names and units attached to each.

ACTION TAKEN: None.

**6. Table 1, section 5.3, Table 5.** *[These] are very useful. The instructions on how to complete attributes vary within a single agency and across agencies, so are confusing to create.*

RESPONSE: Thank you!

ACTION TAKEN: None.

**7. Section 6 (excerpt).** *I wonder how or what the authors think about gradations of flexibility? i.e. what if someone wants to use Celsius instead of K… where do things break down and how much flexibility or personalization can a site do to the MODFs before they have gone off track, before it ceases to be worth the effort of creating it. Is the answer or guiding principle or hierarchy / prioritization of needs that the MODF should faithfully match the MMDF in all ways possible? The more the better? … can the authors comment on the hierarchy of needs or how best to prioritize effort or adherence to the principles or roadmap outlined here. What is the most generalizable golden rule or guiding principle? How will people know whether they are "doing it right" or "doing it wrong", regardless of how much work they are putting in*

RESPONSE: The reviewer raises good points that have been argued philosophically among the 23 coauthors as we have developed the framework for MODFs and MMDFs.

The whole essence of model–observation interoperability is based on having common standards, yet common standards can be limiting for specific purposes.

There are several issues such as (1) unit conversions can result actually changing the data, (2) some data producers can only publish highly processed data sets because of legal risks of misinterpretations based on raw data that has not been QC'ed, and (3) whether data centers have MODF/MMDF "checkers" available as an on-line tool. We have tried in this manuscript to indicate that there is some flexibility in MODFs and MMDFs: we have been open about inventing CMIP-like names when the CMIP6 Table was insufficient to our needs, we have stressed the creative use of attributes like *comment* to add metadata that won't fit under other attributes, and we have used the nominally global attribute *history* in an "off-label" way as a variable attribute.

The other issue is whether there is a realm between "nothing" and "everything", of how to keep the quest for perfect compliance from preventing a good improvement.

ACTION TAKEN: We have tried to further highlight that there is some flexibility in the system. (See Section 6 "We acknowledge", lines 562-567; Discussion "Currently …", lines 531-539; and Section 5's 2nd introductory paragraph, lines 308-314, in revised manuscript). We have also called out the ability of users to contribute to enhancements of the CF Conventions (See Section 5.3 *standard_name* paragraph, lines 439-443, in revised manuscript). And we have added a short paragraph addressing that realm between "nothing" and "everything". (See Section 6 "Creating merged data products …", lines 609-614 in revised manuscript).